# Combination of Xpert® MTB/RIF and Determine™ TB-LAM Ag improves the diagnosis of extrapulmonary tuberculosis at Jimma University Medical Center, Oromia, Ethiopia

**Asnake Simieneh**[1,2,3☯]**, Mulualem Tadesse**[1,2☯]***, Wakjira Kebede**[1,2]**, Mulatu Gashaw**[2]**, Gemeda Abebe**[1,2]

**1** Mycobacteriology Research Center, Jimma University, Jimma, Ethiopia, **2** School of Medical Laboratory Sciences, Faculty of Health Sciences, Jimma University, Jimma, Ethiopia, **3** Department of Medical Laboratory Sciences, Mizan-Tepi University, Mizan, Ethiopia

☯ These authors contributed equally to this work.
* mulualemt.tadesse@gmail.com

**Data Availability Statement:** The datasets used and analyzed during the current study were uploaded as supporting information file (S2 File).

## Abstract

### Background

Ethiopia is one of the high burden countries for extrapulmonary tuberculosis (EPTB); however, the prompt diagnosis of EPTB remains challenging. This study is aimed to evaluate the diagnostic performance of Xpert MTB/RIF and Determine™ TB-LAM Ag (TB-LAM) for the prompt diagnosis of EPTB in Ethiopia.

### Methods

A total of 147 presumptive EPTB patients, including 23 HIV- positive participants were enrolled. Extra-pulmonary samples were collected from all presumptive EPTB cases and tested for *Mycobacterium tuberculosis* complex (MTBC) using fluorescent microscopy, Xpert MTB/RIF, and culture. Additionally, urine samples were also collected from 126 participants and were tested by Determine™ TB-LAM Ag (Alere Inc, Waltham, USA). The Sensitivity and specificity of Xpert and TB- LAM tests were calculated by comparing with a composite reference standard (CRS), which comprises smear microscopy, culture and response to empirical anti-TB treatment.

### Results

Of 147 patients, 23 (15.6%) were confirmed EPTB cases (culture-positive), 14 (9.5%) were probable EPTB (clinically, radiologically or cytologically positive and received anti-TB treatment with good response), and 110 (74.8%) were classified as "non- TB" cases. Compared to the composite reference standard (CRS), the overall sensitivity and specificity of Xpert MTB/RIF were 43.2% and 100%, respectively with the highest sensitivity for Lymph node aspirate (85.7%) and lower sensitivity for pleural fluid (14.3%) and 100% specificity for all specimen types. The sensitivity and specificity of TB-LAM were 33.3% and 94.4%

**Funding:** This work was supported by Jimma University-Mycobacteriology Research Center, Jimma, Ethiopia. The funders had no role in study design, data acquisition, analysis and interpretation, or the decision to prepare the manuscript and submit for publication.

**Competing interests:** NO authors have competing interests.

respectively with the highest sensitivity for HIV co-infected participants (83.3%). The sensitivity of the combination of Xpert MTB/RIF and TB-LAM tests regardless of HIV status was 61.1% whereas the sensitivity was improved to 83.3% for HIV-positive cases.

## Conclusion

TB-LAM alone has low sensitivity for EPTB diagnosis; however, the combination of TB-LAM and Xpert MTB/RIF improves the diagnosis of EPTB particularly for countries with high EPTB and HIV cases.

## Background

Ethiopia is among the 30 highest-burden countries for tuberculosis (TB), TB-HIV coinfection and multidrug-resistant TB (MDR-TB) in the world [1]. In Ethiopia, extrapulmonary TB (EPTB) is very high- accounting for more than 33% of all forms of TB [1, 2]. EPTB contributes significantly to TB-related morbidity and can cause complications, lifelong sequelae, and death. However, the diagnosis of EPTB remains challenging mainly because of the variable non-specific presentations, the paucibacillary nature of the disease, and the difficulty in obtaining appropriate and adequate samples [2–4]. Conventional techniques such as smear microscopy has limited sensitivity for the diagnosis of EPTB, due to the paucibacillary load of the diseases. Mycobacterial culture has long turnaround time and requires bio-safety level III facilities which limits its applicability in resource poor countries [5, 6]. Thus, the diagnosis of EPTB is often made on clinical suspicion alone, resulting in both under- and over-diagnosis and relatively poor outcomes.

Xpert MTB/RIF (Cepheid, Sunnyvale, CA, USA) is a cartridge-based nucleic acid amplification test for the diagnosis of TB and rifampicin resistance and the test has been endorsed by the WHO for use in resource constrained settings [7]. Steingart *et al.* [8] in their Cochrane systematic review reported high diagnostic accuracy (sensitivity 89% and specificity 99%) of Xpert MTB/RIF for pulmonary TB detection. However, Tortoli, *et al* [9] investigated large numbers of consecutive extrapulmonary clinical specimens with Xpert MTB/RIF and obtained a heterogeneous sensitivity across different specimens (sensitivity >85% for CSF, biopsies, urines, pus samples and fine-needle aspirates but very poor sensitivity for the body fluids). More recent study in Ethiopia also reported a heterogeneous sensitivity of Xpert MTB/RIF for the diagnosis of different forms of EPTB (from 30% for pleural TB and up to 90% for lymph node TB) [10]. Thus, more evidence is still needed on the diagnostic performance of Xpert MTB/RIF for the diagnosis of the different forms of EPTB [11].

Lipoarabinomannan (LAM) is an immunogenic lipopolysaccharide which is found in mycobacterial cell walls. It is released from metabolically active or degenerating bacterial cells and present mainly in people with active TB disease and has shown only low cross-reactivity with non-tuberculosis mycobacterial infections [12, 13]. LAM is excreted and detected in urine during advanced immunodeficiency such as HIV coinfection; which results in systemic dissemination of *M. tuberculosis* and high mycobacterial burden due to podocyte dysfunction and change in glomerular filtration rate that leads higher antigen concentration of LAM in urine [14, 15]. The other possible way for LAM excretion in urine is due to renal involvement of patients with disseminated TB in patients living with HIV and advanced immunodeficiency [16].

Most laboratory diagnostic tests of EPTB involves invasive procedures to obtain specimens, which further complicates the diagnostic protocol [17]. Thus, there is a need for rapid point of

care and non-invasive tests for the diagnosis of EPTB. Determine[TM] TB LAM Ag (TB-LAM) detects LAM in the patient's urine for the diagnosis of TB. Urine-based testing would have advantages over other specimen based testing because urine is easy to collect and store, and lacks the infection control risks associated with collection. WHO recommends TB-LAM for people living with HIV with symptoms suggestive of pulmonary and/or extrapulmonary TB, with CD4 ≤100 cells/mm$^3$ or in those seriously ill [13]. However, the yield of Xpert MTB/RIF and TB-LAM tests used in combination for the diagnosis of EPTB is unknown. This study is aimed to evaluate the diagnostic performance of Xpert MTB/RIF and TB-LAM for the diagnosis of EPTB in Ethiopia.

## Materials and methods

### Study subjects, sample collection and laboratory tests

This study was conducted at Jimma University Medical Center, Southwest Ethiopia from April to October 2019. A total of 147 presumptive EPTB patients were consecutively enrolled in the study. Consecutive patients were included based on clinical suspicion of EPTB. As part of routine clinical practice, body fluids such as cerebrospinal fluid, pleural fluid, peritoneal fluid, pericardial fluid, synovial fluid, and lymph node aspirate were collected according to standard operating procedure (SOP). Gastric aspirate was collected early in the morning through nasogastric tube following an overnight fast. Each sample was divided into two and transferred into falcon tubes. The first sample was processed for Xpert MTB/RIF at Jimma University Medical Center-Microbiology Laboratory and the second sample was transported to Jimma University-Mycobacteriology Research Center where smear microscopy and L-J culture were done. In addition, all presumptive EPTB patients were requested to submit urine specimen (10-30ml) for the research purpose. Immediately after collection, urine specimen was transported to the Mycobacteriology Research Center and stored at -20˚c until processing.

Demographic and clinical characteristics were gathered through interview by using a pre-tested questionnaire. Participants' medical records were reviewed for HIV status, HIV-WHO staging, clinician's decision to initiate anti-TB treatment (ATT) and response to anti-TB treatment.

### Mycobacterial culture

Culture was done on Lowenstein Jensen (L-J) medium. Lymph node aspirate, gastric aspirate and blood-stained specimens were decontaminated by the standard *N*-acetyl-L-cysteine and sodium hydroxide (NALC/NaOH) method with a final NaOH concentration of 1% [18]. An equal volume of standard NALC/NaOH solution was added to the specimen and incubated for 15–20 minutes. After centrifugation for 15 minutes at 3000g, the sediment was resuspended in 1.0 to 1.5 ml of sterile phosphate buffered saline (pH 6.8). Specimens expected to be sterile (such as cerebrospinal fluid (CSF), pleural fluid, and peritoneal fluid) were directly centrifuged to concentrate the samples. L-J tubes were inoculated with 0.2 mL (2–4 drops) of the processed specimens [19]. For positive L-J culture, a smear was prepared to detect acid-fast bacilli (AFB), and *Mycobacterium tuberculosis* complex (MTBC) was confirmed by a p-nitro benzoic acid inhibition test [19].

### Fluorescent smear microscopy

Smears were prepared from all processed samples other than urine and examined at Mycobacteriology Research Center of Jimma University (JU-MRC). Approximately 10μl of the pellet was deposited on a clean microscope slide and spread using the side of the pipette tip. Smears

were air-dried and heat-fixed and stained with the Auramine-phenol method according to the SOP. The stained smears were examined under the light-emitting diode fluorescent microscopy (Primo Star iLED, Carl Zeiss, Gottingen, Germany) with 200x and 400x magnification for AFB [19].

## Xpert MTB/RIF test

The Xpert MTB/RIF assay was performed as previously described by Helb *et al.* and WHO [20, 21]. Briefly, sample reagent was added in a 2:1 ratio to patient specimen. The mixture was vortexed and incubated at room temperature for 15 minutes. Two ml of the reagent sample mixture was transferred to an Xpert cartridge using a Pasteur pipette. Then the cartridge was loaded onto Xpert machine (GeneXpert- Dx System version 4.4a, Cepheid Company, 904 Caribbean Drive, CA 940889, USA) and results was automatically generated after 1 hour and 50 minutes.

## TB-LAM

The stored urine specimen in a deep freeze was de-frozen and processed as previously described [13, 22, 23]. About 10-30m ml of the de-frozen urine sample was centrifuged at 10,000g for 5 minutes, and 60μl of the supernatant was applied to the TB-LAM test strip [Determine$^{TM}$ TB-LAM Ag test; Abbott, Waltham, MA (formerly Alere)] on the same day with results interpreted using a 4-grade scale, with grade one or above constituting a positive result. The strip was visually inspected by 2 trained study staff members exactly at the end of 25 minutes incubation period. Those individuals who interpreted the TB-LAM were blinded to patients' clinical data and TB diagnostic status. Any disagreement between interpreters were decided by the decision of a third interpreter. TB-LAM results were not used for patient management or treatment initiation.

## Diagnostic classification for analysis

Based on clinical and laboratory findings, study participants were categorized as follows: (i) Confirmed TB: defined as a positive culture of MTBC and /or smear positive; (ii) Probable TB: culture negative but clinical improvement after anti-TB treatment (ATT); (iii) Non TB: patients for whom no microbiological evidence of TB (smear-negative and culture-negative), and/or for whom an alternative diagnosis is available. Composite reference standard (CRS), which comprises smear microscopy, L-J culture and clinical improvement after ATT initiation, was used as gold standard to calculate sensitivity, specificity and predictive values. Any patient that was positive for any one component of the CRS was considered as 'TB' cases.

## Data management and analysis

Data were entered through Epidata version 3.1 and analyzed using SPSS software package version 20. The sensitivity, specificity, positive predictive value (PPV) and negative predictive value (NPV) of TB-LAM and Xpert MTB/RIF test were calculated in comparison with culture alone and with CRS. The results of the two TB-LAM test readers were compared, and the inter observer agreement was calculated using the Cohen's kappa coefficient (κ) statistic. Overlapping 95% CI data were considered as showing no significant difference between test results. Comparison of proportion between the methods was done by Chi square test using the MedCalc Software (https://www.medcalc.org/calc/comparison_of_proportions.php) and the *P*-value < 0.05 showed statistically significance difference between the methods.

### Ethics approval and consent to participate

This study was approved by institutional review boards of Jimma University, Ethiopia (Ref. No. IHRPGD/389/2019). Written informed consent was obtained from all participants for use of routine clinical data for research purposes. Laboratory results were reported back to the physicians for treatment initiation or decision as early as available.

## Results

### Demographic and clinical characteristics

A total of 147 presumptive EPTB patients were included. Participants had a median age of 35 years (IQR, 22–45) and 82 (56%) were males. The majority, 84 (57.1%) of the participants were from rural areas (**Table 1**). Regarding clinical signs and symptoms, 124 (85%) and 122 (83%) of the study participants had tiredness and loss of appetite, respectively. Among the 134 (91.2%) presumptive EPTB cases with documented HIV status, 23 (15.6%) were HIV positive, and most (89.9%) of them were already on antiretroviral therapy (ART) for a mean of 8 years (SD ±1.92).

All participants provided sufficient volume of site specific extrapulmonary specimens including 49 peritoneal fluid, 45 pleural fluids, 28 CSF, 19 lymph node aspirates, 2 pericardial and 4 other fluids (2 gastric aspirates and 2 synovial fluids) (**Table 1**). In addition, all study participants were requested to provide urine specimen but only 126 (86%) of study participants provided urine specimen (**Fig 1**).

### Diagnosis of EPTB

In total, 147 extrapulmonary samples were successfully analyzed by fluorescent microscopy, Xpert MTB/RIF, and L-J culture. Of 147 presumptive EPTB cases, 49 (33.3%) were suspected for abdominal TB and 45 (30.6%) for pleura TB. Fifty-three (36.1%) of presumptive EPTB

**Table 1. Socio demographic and clinical characteristics of presumptive EPTB cases visited Jimma University Medical Center from April to October 2019(N = 147).**

| Characteristics | | All,(N = 147) | Confirmed TB(n = 23) | Probable TB (n = 14) | Non TB (n = 110) |
|---|---|---|---|---|---|
| Sex | Male | 82(55.8%) | 10(43.5%) | 11(78.6%) | 61(55.5%) |
| | Female | 65(44.2%) | 13(56.5%) | 3(21.4%) | 49(44.5%) |
| Residence | Urban | 63(42.9%) | 12(52.255) | 6(42.9%) | 45(40.9%) |
| | Rural | 84(57.1%) | 11(47.8%) | 8(57.1%) | 65(59.1%) |
| Age (years) | 0–15 | 24(16.3%) | 4(17.4%) | 2(14.3%) | 18(16.4%) |
| | 16–30 | 35(23.8%) | 10(43.5%) | 3(21.4%) | 22(20%) |
| | 31–45 | 52(35.3%) | 6(26.1%) | 4(28.6%) | 42(38.2%) |
| | >45 | 36(24.5%) | 3(13.0%) | 5(35.7%) | 28(25.5%) |
| Types of specimen | Peritoneal fluid | 49(33.3%) | 2(8.7%) | 6(42.9%) | 41(37.3%) |
| | Pleural fluid | 45(30.6%) | 4(17.4%) | 3(21.4%) | 38(34.5%) |
| | Cerebrospinal fluid | 28(19%) | 3(13%) | 3(21.4%) | 22(20%) |
| | Lymph node aspirate | 19(12.9%) | 13(56.5%) | 1(7.1%) | 5(4.5%) |
| | Pericardial fluid | 2(1.4%) | 0 | 0 | 2(1.8%) |
| | Others* | 4(2.7%) | 1(4.3%) | 1(7.1%) | 2(1.8%) |
| HIV status | Positive | 23(15.6%) | 4(17.4%) | 2(14.3%) | 17(15.5%) |
| | Negative | 111(75.5%) | 16(69.6%) | 12(85.7%) | 83(75.5%) |
| | Unknown | 13(8.8%) | 3(13%) | 0 | 10(9.1%) |

*Others include 2 gastric aspirates and 2 synovial fluid specimens. Of these, 1 gastric aspirate specimen was positive on L-J culture.

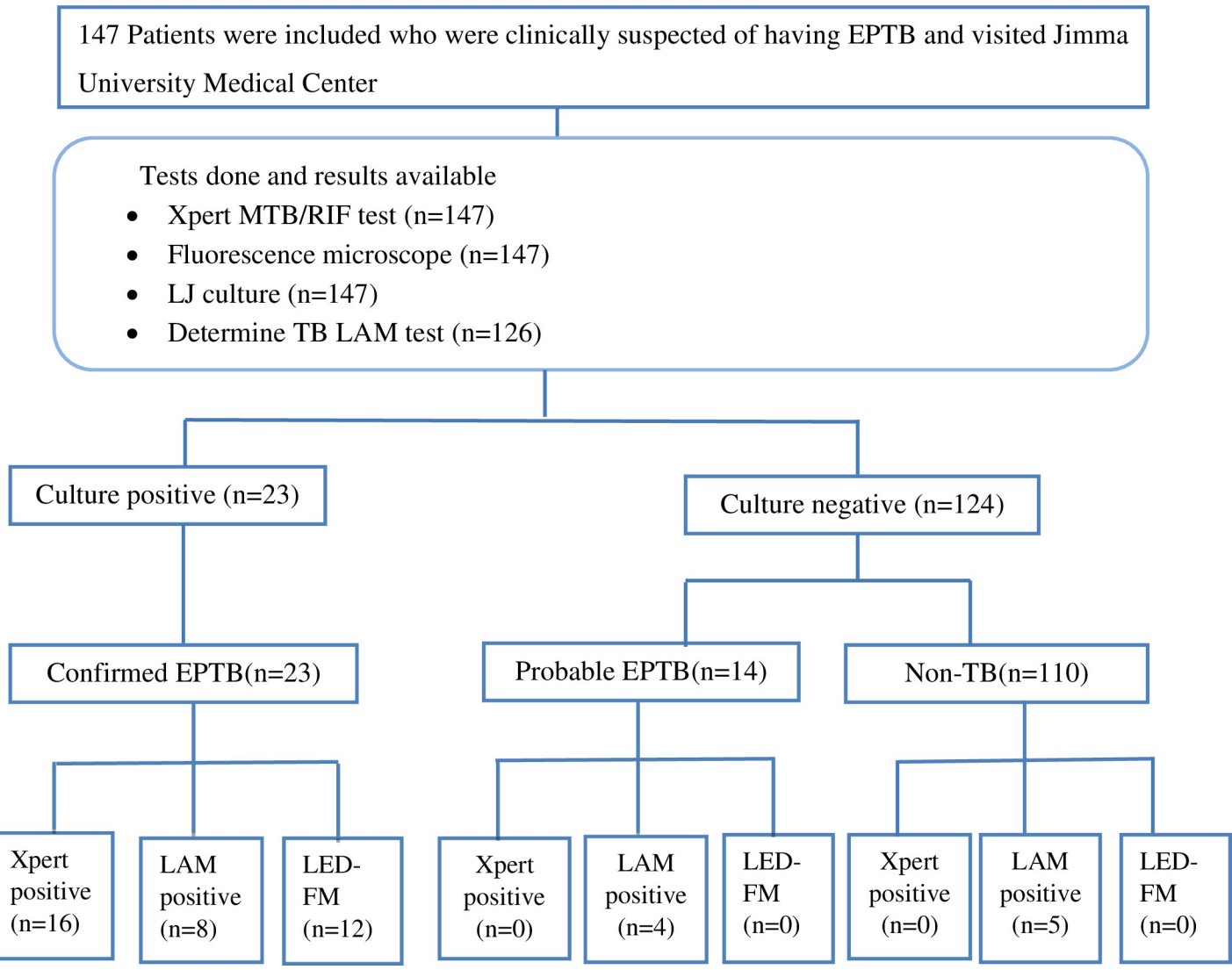

**Fig 1. Flow chart showing overall patient flows, diagnostic test result and diagnostic classifications.**

cases had various sites affected, including meningitis, lymph node, skin, and pericardial. However, TB disease detection (confirmed cases) rate was highest among patients suspected of having lymph node TB (56.5%) followed by pleura TB (17.4%) and TB meningitis (13%).

Bacteriologically confirmed EPTB was found in 23 of the presumptive EPTB cases, for a detection rate of definite EPTB of 15.6% (95% CI: 10.2%, 22.5%). Of these, 13 had lymph node TB, 4 had TB pleuritis, 3 had TB meningitis and 2 had peritoneal TB (**Table 1**). One TB case was identified from gastric aspirate from whom pulmonary manifestation was excluded (**Table 1**). Among 23 culture-positive EPTB cases, 16 (69.6%) were Xpert MTB/RIF positive, with no rifampicin resistance detected and 8 (34.8%) were positive for AFB by fluorescent microscopy. Furthermore, there were 14 (11.3%) patients with probable TB- cases with clinical improvement after clinician's decision to start ATT whereas 110 patients were classified as 'non TB' cases because no bacteriological or clinical evidence for TB was observed (**Fig 1**). The combined detection rate of definite and probable EPTB was 25.2% (95% CI: 18.4–33.0); 37/ 147. Compared to CRS (definite and probable EPTB cases), L-J culture had a sensitivity of

**Table 2. Diagnostic performance of TB-LAM and Xpert MTB/RIF test.**

| Diagnostic Accuracy | Culture as a reference standard | | |
|---|---|---|---|
| | TB-LAM test | Xpert MTB/RIF test | Combination of TB-LAM and Xpert MTB/RIF tests |
| Sensitivity (95%CI) | 34.8% (16.4–57.3);8/23 | 69.6% (47.1–86.8);16/23 | 82.6% (61.2–95.1);19/23 |
| Specificity (95%CI) | 91.3% (84.1–95.9);94/103 | 100% (97.1–100);124/124 | 92.2% (85.3–96.6);95/103 |
| PPV (95%CI) | 47.1% (27.8–67.3);8/17 | 100%(79.4–100); 16/16 | 70.4% (54.3–82.6);19/27 |
| NPV (95%CI) | 86.2% (82.2–89.5);94/109 | 94.7% (90.5–97.1);124/131 | 96. % (90.7–98.3);95/99 |
| | CRS as a reference standard | | |
| Diagnostic Accuracy | TB-LAM test | Xpert MTB/RIF test | Combination of TB-LAM and Xpert MTB/RIF tests |
| Sensitivity (95%CI) | 33.3% (18.6–51);12/36 | 43.2% (27.1–60.5);16/37 | 61.1% (43.5–76.9);22/36 |
| Specificity (95%CI) | 94.4% (87.5–98.2);85/90 | 100% (96.7–100);110/110 | 94.4% (87.5–98.2);85/90 |
| PPV (95%CI) | 70.6% (47.7–86.4);12/17 | 100% (79.4–100);16/16 | 81.5% (64.4–91.5);22/27 |
| NPV (95%CI) | 78% (73.7–81.8);85/109 | 84% (79.8–87.4);110/131 | 85.9% (80.1–90.2);85/99 |

**Key:** PPV; Positive predictive value, NPV; Negative predictive value.

62.2% (23/37). As shown in **Table 2**, for determination of sensitivity, TB-LAM test was compared against 36 and Xpert MTB/RIF test against 37 positive samples found by CRS.

## Diagnostic performance of Xpert MTB/RIF

The diagnostic accuracy of Xpert MTB/RIF test to diagnose EPTB regardless of patients' HIV status is shown in **Table 2**. Using L-J culture as a comparator, the sensitivity and specificity of Xpert MTB/RIF were 69.6% (95%CI, 47.1–86.8); 16/23 and 100% (95%CI, 97.1–100); 124/124, respectively. However, the sensitivity was reduced to 43.2% (95%CI, 27.1–60.5); 16/37 when compared to CRS without affecting the specificity. When stratified by HIV status, the Xpert MTB/RIF sensitivity was higher in HIV positive patients; 50% (95%CI, 11.8–88.2); 3/6 compared to HIV negatives 35.7% (95%CI, 18.6–56); 10/28 (**Fig 2**). The diagnostic accuracy of Xpert MTB/RIF test among different specimens were also determined. The sensitivities among the specimen types differed markedly. Comparing to CRS, the sensitivity of Xpert MTB/RIF was 85.7% (95%CI, 57.2–98.2); 12/14 for lymph node aspirate specimen, 33.3% (95%CI, 4.3–77.7); 2/6 for CSF and only 14.2% (95%CI, 0.4–57.9); 1/7 for pleural fluids (**Table 3**). Xpert MTB/RIF on site specific samples had higher sensitivity 44.4% (95%CI, 28–62); 16/36 than urine TB-LAM 33.3% (95%CI, 18.6–51); 12/36 performed for the same patient (n = 126), p<0.001.

## Diagnostic performance of TB-LAM

Out of 126 presumptive EPTB cases submitted urine sample for the TB-LAM, 17 (13.5%) had a positive TB-LAM result. Among 23 patients with culture-confirmed EPTB, 8 (34.8%) had a positive TB-LAM, and 4 (28.6%) TB-LAM positive cases were among probable EPTB patients (negative-Xpert MTB/RIF and -culture results). There were 5 cases with positive TB-LAM result but no evidence of TB (bacteriologically and clinically negative for TB) (**Fig 1**).

When compared to CRS, the TB-LAM had lower sensitivity (33.3%; 95%CI, 18.6–51); 12/36 whereas higher specificity performance (94.4%; 95%CI, 87.5–98.2); 85/90. There is a significant difference in sensitivity of TB-LAM assay in HIV-positive and HIV-negative participants, which was 83.3% (95%CI, 35.9–99.6); 5/6 in HIV-positives and 26% (95%CI, 11.1–46.3); 7/27 in HIV-negatives (*P*-value = 0.0094) (**Fig 2**).

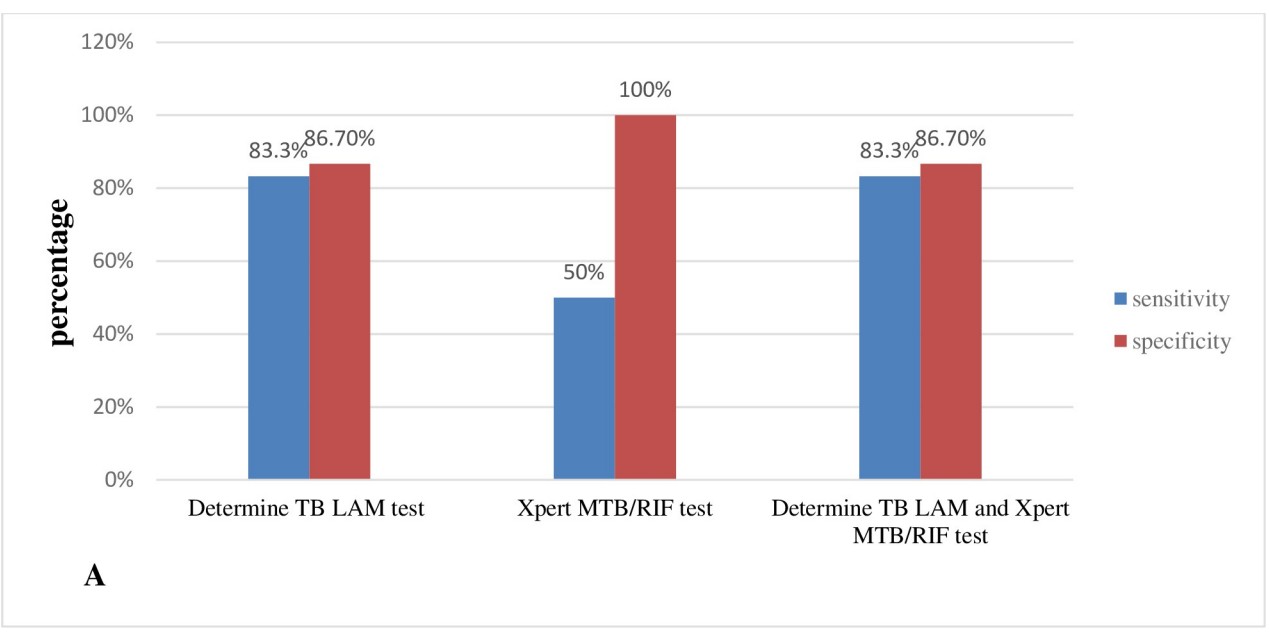

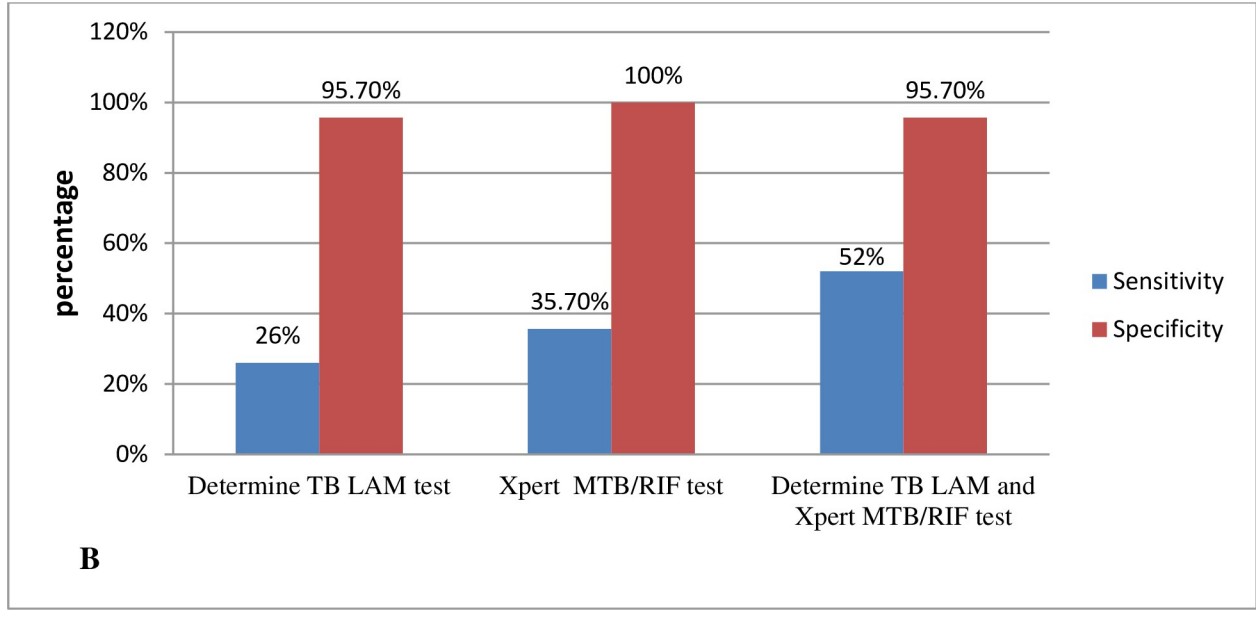

**Fig 2.** Performances of TB-LAM, Xpert MTB/RIF and combination of TB-LAM and Xpert MTB/RIF tests comparing to CRS among HIV positive (A) and HIV negative (B) cases.

## Combination of Xpert MTB/RIF and TB-LAM

The combined use of TB- LAM and Xpert MTB/RIF tests had the sensitivity of 61.1% when compared to CRS, which was significantly higher than the sensitivity of each test alone (**Table 2**). Most importantly, in stratified analysis, combining the two tests (Xpert MTB/RIF and TB-LAM) have significantly improved the sensitivity to 83.3% in HIV-positives compared to HIV-negative patients (52%) (**Fig 2**). The detection rate and diagnostic performance of

**Table 3. Performance of Xpert MTB/RIF test among different non respiratory samples.**

| Sample type | Total | Xpert MTB/RIF | | CRS | | Sensitivity (95%CI) | Specificity (95%CI) |
|---|---|---|---|---|---|---|---|
| | | Positive | Negative | Positive | Negative | | |
| Lymph node | 19 | 12 | 7 | 14 | 5 | 85.7% (57.2–98.2);12/14 | 100% (47.8–100);5/5 |
| CSF | 28 | 2 | 26 | 6 | 22 | 33.3% (4.3–77.7);2/6 | 100%(84.6–100);22/22 |
| Pleural fluid | 45 | 1 | 44 | 7 | 38 | 14.3% (0.4–57.9);1/7 | 100%(90.6–100);38/38 |
| Other* | 4 | 1 | 3 | 2 | 2 | - | - |

**Note:** other* includes (2 gastric aspirate and 2 synovial fluid). All peritoneal fluid and pericardial fluid specimens were negative for Xpert MTB/RIF test.
**Key:** CRS; Composite reference standard.

TB-LAM, Xpert MTB/RIF and combination of TB-LAM and Xpert MTB/RIF in HIV-positive and HIV-negative participants was described more in supporting file (**S1 File**).

## Discussion

Sensitive and rapid diagnostic tests will increase the number of diagnosed patients; facilitate early treatment, contributing potentially to decreased disease transmission and reduced case fatality, especially in the highly lethal form of TB (such as disseminated TB). Conventional techniques (smear microscopy and culture) have limited sensitivity for the diagnosis of EPTB, due to the paucibacillary load of the diseases. *M. tuberculosis* needs about 6–8 weeks to grow on Lowenstein Jensen (L-J) culture, which limits the applicability of culture for immediate case management. In most cases, it is difficult to obtain appropriate extrapulmonary specimen and there is a need for invasive procedures. Incorporating tests like TB-LAM which uses urine sample for TB diagnosis is attractive than the other EPTB diagnostic tools which need invasive procedures for sample collection [13, 24]. This study evaluated the performance of two rapid point-of-care tests (Xpert MTB/RIF and TB-LAM) for the diagnosis of EPTB in Ethiopia.

In our study, we found variable sensitivity of Xpert MTB/RIF among different extrapulmonary specimen types. We documented the higher sensitivity of Xpert MTB/RIF in lymph node aspirates and lower sensitivity in CSF and pleural fluid specimens; which is consistent with findings of previous studies [10, 25, 26]. The poor sensitivity in such fluids is probably due to the paucibacillary nature of the disease. Other studies suggested Xpert MTB/RIF Ultra over the conventional Xpert MTB/RIF with an overall improvement of 45.6% for the diagnosis of respiratory and non-respiratory smear-negative cases and paucibacillary diseases [27]. This is mainly because Xpert MTB/RIF Ultra has lower analytical limit of detection than Xpert MTB/RIF (15 CFU/mL versus 100–120 CFU/mL respectively) [28].

Although Xpert MTB/RIF has lower sensitivity than culture, it has some advantages over conventional culture. Xpert MTB/RIF has shorter turnaround time for the diagnosis of TB especially when a rapid diagnosis of TB and early treatment initiation is significantly necessary, particularly in highly lethal form of TB such as TB meningitis. Xpert MTB/RIF can simultaneously detect *M. tuberculosis* and rifampicin resistance within 2 hours in contrast to L-J culture that could take 2 to 6 weeks for *M. tuberculosis* to grow and conventional drug resistance tests could take 3 extra weeks. As a result Xpert MTB/RIF provides the information which is used to select treatment alternatives and for early infection control of drug resistant TB [29].

The other interesting finding from the current study is that the high specificity and positive predictive value of Xpert MTB/RIF, which provides significant evidences for the clinicians to rule in EPTB and to treat patients as early as possible following a positive Xpert MTB/RIF result. Whereas the overall low sensitivity of Xpert MTB/RIF (43.2%) compared to CRS shows that negative Xpert MTB/RIF result does not rule out EPTB diagnosis especially from body

fluid specimens such as pleural and peritoneal fluids. In the future, the implementation of Xpert MTB/RIF Ultra has a potential to overcome the sensitivity limitations of Xpert MTB/RIF in pleural or peritoneal specimens [27, 30].

Overall, the sensitivity performance of TB-LAM for EPTB is low in our study. The reported performance of the TB-LAM varied greatly among different published studies [22, 31, 32]. A systematic review on TB-LAM for the diagnosis of pulmonary TB using microbiological reference standard showed a pooled sensitivity of 42% and specificity of 91% [33], which has been found higher sensitivity compared with the sensitivity of the current study, with no observed difference of specificity (94.4%). This low sensitivity of TB-LAM indicates that negative TB-LAM result cannot be used to rule out EPTB. However, TB-LAM alone should not be used to diagnose EPTB. Although low sensitivity of urine TB-LAM compromises its utility as a replacement for culture or Xpert MTB/RIF on respiratory samples, the ease of access of urine (easily obtained from children and adults), processing and storage and the low infection risk to health professionals during urine collection makes TB-LAM attractive. Additionally, TB-LAM can be done anywhere, including at the patient's bedside, and takes only 25 minutes to perform [34].

In our study, there were 5 TB-LAM positive cases for whom TB was bacteriological and clinically ruled out. Absence of bacteriologically confirmed or clinically suggestive-TB strongly supports that these were indeed false positive results. In some cases, factors other than active TB are likely to be responsible for positive urine TB-LAM results. For instance, interference by non-tuberculous mycobacteria or other bacterial species, as well as contamination from environment or stool could be a cause for false positive result [35]. The other possible cause could be variation in the interpretation of the TB-LAM result. In our study, grade 1 cut-off point (the band of lowest intensity) was used as a threshold. Peter *et al*. found decreased specificity when the grade 1 cut-off point was used as threshold, as compared to using the grade 2 cut-off point [36].

In our subgroup analysis, the sensitivity of TB-LAM in HIV positive patients was much higher than in HIV negative patients. In this study TB-LAM detected TB in additional 3 and 2 HIV infected patients who were missed by smear microscopy and Xpert MTB/RIF respectively. This suggests that TB-LAM could somehow address the challenges for diagnosing TB in HIV co-infected patients through smear microscopy and Xpert MTB RIF test. The higher sensitivity of TB-LAM in HIV positive patients than HIV negative patients might be due to podocyte dysfunction that results change in glomerular filtration which is most of the time secondary to HIV disease that leads higher antigen concentration of LAM in urine [14]. Although not assessed in our study, different studies shows a significant association between the performances of TB-LAM and level of immunosuppression [37–39] and positive TB-LAM is as a predictor for mortality [22, 39]. In their recent study, Kerkhoff *et al*. [40] reported substantially higher sensitivity of Fujifilm SILVAMP TB-LAM (FujiLAM) over TB-LAM for detecting EPTB in HIV inpatients with moderate sensitivity in pleural fluid and CSF. This suggests a potential use of FujiLAM as a first-line test for the rapid detection of EPTB in HIV patients, with substantial added benefit in paucibacillary diseases such as pleural TB and TB meningitis.

The overall sensitivity of the combination of Xpert MTB/RIF and TB-LAM irrespective of HIV status was higher than either test alone in this study, which was 61.1% and it is comparable to L-J culture 62.2%. Most importantly, the sensitivity was increased to 83.3% when these two tests (Xpert MTB/RIF and TB-LAM) were used together in HIV-positive patients. This finding suggests that it is better to use the combination of Xpert and TB-LAM tests than each test alone or culture alone when diagnosing EPTB. Both Xpert and TB-LAM have short turnaround time than culture which results in early diagnosis and treatment of TB patients, which

is one of the main pillars and components of WHO End TB strategies [24]. It is noticeable that using the combination of Xpert and TB-LAM is relatively more expensive than using each test alone, however it provides important contribution for the diagnosis of EPTB especially when there is a need for very sensitive test and short time to treatment initiation as in the case of disseminated TB, TB meningitis or TB in HIV infected patients. However, patients whose ultrasound and radiological or clinical findings strongly suggests EPTB should start ATT despite a negative Xpert MTB/RIF and TB-LAM results.

Our study is not without limitations. First, since some patients were not able to provide urine specimen, they had no TB-LAM results available for analysis. Second, the use of frozen urine samples for TB-LAM would have affect the sensitivity. Though not reproduced by other investigators, Peter *et al.* has reported that the use of frozen urine has been associated with reduced TB-LAM sensitivity [36].

## Conclusions

Xpert MTB/RIF has the highest sensitivity for lymph node aspirates and lowest sensitivity for pleural fluids. The overall sensitivity of TB-LAM was low, but the sensitivity was significantly improved among EPTB patients infected with HIV. The sensitivity of combination of Xpert MTB/RIF and TB-LAM was superior to either test alone or equivalent to culture to diagnose EPTB. The evidence from this study suggests that the combination of these tests can improve the diagnosis of EPTB particularly in patients infected with HIV.

## Supporting information

**S1 File. Detection rate and the diagnostic performance of TB-LAM, Xpert MTB/RIF and combination of TB-LAM and Xpert MTB/RIF in HIV-positive and HIV-negative participants.** (RTF)

**S2 File. The original SPSS dataset used and analyzed in the current study.** (SAV)

## Acknowledgments

We are grateful to the study participates who consented to take part in this study. We would also like to thank the staff of Mycobacteriology Research Center and laboratory personnel at Jimma University Hospital Laboratory for the assistance and guidance during laboratory work and data collection.

## Author Contributions

**Conceptualization:** Asnake Simieneh, Mulualem Tadesse.

**Data curation:** Asnake Simieneh, Wakjira Kebede.

**Formal analysis:** Asnake Simieneh, Wakjira Kebede.

**Investigation:** Mulualem Tadesse, Gemeda Abebe.

**Methodology:** Asnake Simieneh, Mulualem Tadesse.

**Project administration:** Gemeda Abebe.

**Supervision:** Mulualem Tadesse, Mulatu Gashaw, Gemeda Abebe.

**Writing – original draft:** Asnake Simieneh, Wakjira Kebede.

**Writing – review & editing:** Mulualem Tadesse, Mulatu Gashaw, Gemeda Abebe.

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
