## [Decision Letter · Decision Letter 0]

18 Jun 2021

PONE-D-21-10435

Combination of Xpert®MTB/RIF and DetermineTMTB LAM assayimproves the diagnosis of extrapulmonary tuberculosis at Jimma University Medical Center, Oromia, Ethiopia

PLOS ONE

Dear Dr. Tadesse,

Thank you for submitting your manuscript to PLOS ONE. After careful consideration, we feel that it has merit but does not fully meet PLOS ONE’s publication criteria as it currently stands. Therefore, we invite you to submit a revised version of the manuscript that addresses the points raised during the review process.

We look forward to receiving your revised manuscript.

Kind regards,

Shampa Anupurba, MD

Academic Editor

PLOS ONE

2. Please provide additional details regarding participant consent. In the ethics statement in the Methods and online submission information, please ensure that you have specified whether consent was informed.

**a)** If there are ethical or legal restrictions on sharing a de-identified data set, please explain them in detail (e.g., data contain potentially sensitive information, data are owned by a third-party organization, etc.) and who has imposed them (e.g., an ethics committee). Please also provide contact information for a data access committee, ethics committee, or other institutional body to which data requests may be sent.

Additional Editor Comments (if provided):

The manuscript has been reviewed thoroughly. I have very minor additional comments although they might be repetition of the reviewer's comments:

1. Line 188- 'Forty-nine (33.3%) of the presumptive EPTB cases had the classification of colitis'-On what basis was this classification made and what was the sample collected from these patients?

2. Although 4 of the 23 HIV positive patients were bacteriologically confirmed, it is not clear how many were positive by Xpert MTB/RIF. What is meant by 50% in Line 211? Similarly for Determine TB LAM assay

3. Line 320- How was the sensitivity of LJ culture derived to be 62.2%? What was the gold standard?I think that was not within the scope of the study

4. Fig 1- The flow chart could start with inclusion of 147 patients suspected of presumptive TB instead of 160 patients

Reviewers' comments:

Reviewer's Responses to Questions

**Comments to the Author**

1. Is the manuscript technically sound, and do the data support the conclusions?

Reviewer #1: Yes

2. Has the statistical analysis been performed appropriately and rigorously? 

Reviewer #1: No

3. Have the authors made all data underlying the findings in their manuscript fully available?

Reviewer #1: Yes

4. Is the manuscript presented in an intelligible fashion and written in standard English?

Reviewer #1: No

5. Review Comments to the Author

Reviewer #1: In the current study the authors aim to investigate the diagnostic performance of combined use of Xpert MTB/RIF and TB LAM Ag test for detection of M. tuberculosis complex among EPTB patients in a medical center of Ethiopia. Considering the high prevalence of EPTB and association of HIV in Ethiopia, the current diagnostic approach provides the important insight for increased detection of EPTB cases in clinical settings. However, the current manuscript needs to address several issues before approval.

Major comments:

1. According to the manufacturer (Abbot Laboratories, Chicago, US), the test name is ‘Determine™ TB-LAM Ag’. It is suggested to use the exact test name in the Title of the manuscript. Throughout the text, authors should be consistent mentioning the test name. Examples of some discrepancies are: ‘Determine TB LAM test’ in Line-31, ‘TB LAM’ in Line 41, ‘Determine TB-LAM test’ in line 136, ‘TB-LAM assay’ in line 140 and so on. It is suggested to use ‘Determine™ TB-LAM Ag (TB-LAM)’ when appears first in the text, followed by ‘TB-LAM’ all through the text.

2. In the introduction section, please add one paragraph describing the other published data on the performances of the Xpert MTB/RIF assay and Determine TB-LAM assay for detection of EPTB cases.

3. One of the EPTB samples as has been described inconsistently throughout the text. For example, in Table-1 as ‘pus (abscess)’, Table-2 and throughout the text as ‘Abscess’. According to Line-191, it appears that samples were obtained from ‘Lymph node TB’. It would be easier for the readers to understand if the term ‘Lymph node aspirate’ is used all through the manuscript instead of ‘Abscess’.

4. Gastric aspirate is considered as an alternative specimen for detection of pulmonary TB when patients cannot expectorate the sputum. As this study aims to detect the EPTB cases, authors should exclude the two ‘gastric aspirate’ samples from the study and re-analyze the results and correct the statistics accordingly in different relevant Tables, Figures and throughout the text.

5. For Table-2 and Table-3: Authors should check carefully that the statistics are presented correctly. It seems that the value of sensitivity, specificity, PPV and NPV and their corresponding 95% CI are not accurate. As for example, in Table-2 (based on the calculation of Fig-1), for ‘Determine TB LAM test’: the sensitivity would be 34.8% with 95% CI ranging from 16.4-57.3, specificity would be 92.7% (86.7-96.6), PPV would be 47.1% (27.7-67.4), and NPV would be 88.5% (85.0-91.2). Please check both table 2 and 3, and correct accordingly. Please also add the 95% CI for specificity of Xpert MTB/RIF test in both tables. Should use the value of 95% CI in a consistent format like 0.0% (0.0-0.0) instead of 0.0% (0.0%-0.0%) in the tables and text.

6. For clarity and better understanding authors should include the number of samples detected or not detected by the index tests compared to the reference tests along with the value of sensitivity and specificity. As for example, in Table-2, for ‘Determine TB LAM test’ please write the Sensitivity: 34.8% (34.1-35.5); 8/23 instead of 34.8% (34.1-35.5), and for specificity: 92.7% (86.7-96.6); 115/124 instead of 92.7% (86.7-96.6). Please add the numbers in Table-2 and Table-3 and describe accordingly in the text.

7. Figure 2: please describe in the result section (by mentioning the numbers of samples) how the percentages of different tests were obtained between HIV positive and negative patients.

8. Line 215-217: It is not clear how the sensitivities of 44.4% for Xpert and 30.6% for TB LAM were obtained. Please clarify. Please also mention the statistical methods in ‘Methods and Material’ section that was used to obtain the p value.

9. Recently, Xpert MTB/RIF Ultra and Fujifilm SILVAMP TB-LAM have been appeared with higher sensitivities for detection of tuberculosis. In the discussion section authors should add one paragraph about the future applicability of these two assays compared with the current findings of Xpert MTB/RIF assay and Determine TB LAM assay.

10. Line 344: Please clarify the statement ‘equivalent to culture to diagnose EPTB’. It is not clear how the sensitivity of combined use of Xpert MTB/RIF and TB LAM was equivalent to ‘culture’ as the authors did not mention anything about the sensitivity of ‘culture’ in the manuscript.

11. While the text is readable, there are some grammar mistakes. Please correct the text to improve the comprehensibility.

Minor comments:

Line 37-The authors should mention the name of ‘reference’ test that is the ‘CRS’ to whom the results of Xpert MTB/RIF were compared.

Line 42: Please write ‘EPTB cases’ instead of ‘TB cases’.

Line 41-42: Please correct and re-phrase the sentence ‘The combination of Xpert MTB/RIF and TB LAM detected 61.1% of all EPTB participants and 83.3% of HIV co-infected TB cases’ by mentioning the sensitivity instead of detection.

Line 98: Should write Lowenstein Jensen (L-J) as it appears first in the text.

Line 135: Please mention the volume of urine used for centrifugation.

Authors should specify why the LAM test was performed on refrigerated urine samples instead of freshly collected samples. Should discuss the point in the discussion section whether the sensitivity of LAM test varies between refrigerated versus fresh urine sample.

Line172: Please write as 35 years (IQR, 22-45).

Line-187: ‘Forty-nine (33.3%) of the presumptive EPTB cases had the classification of colitis’. Please clarify whether all of the 49 cases from where peritoneal fluid were collected had ‘colitis’.

Line-195 and 202: Authors should mention ‘detection rate’ instead of ‘prevalence’.

Authors have mentioned in Table-1 that one sample among ‘Others’ group (2 gastric aspirates and 2 synovial fluids) was diagnosed as ‘Confirmed TB’. Please, specifically mention the sample type (either the gastric aspirate or synovial fluid) that was positive for TB in the table and add the description in line-196.

Line 228-229: please write as (0.0%; 95% CI, 0.0-0.0) instead of ‘(30.6% (95%CI, 30%-

229 31%))’ and ‘(93.3% (95%CI, 93.2%-93.5%))’.

Line 230 and 238: Authors should include the ‘p value’ of the significant difference, and the test used.

Line 235-236: Please rephrase the sentence ‘Adding Determine TB LAM test to Xpert MTB/RIF test increases the sensitivity of Xpert MTB/RIF test from 43.2% to 61.1%’ as it does not increase the sensitivity of Xpert MTB/RIF test, but increases the sensitivity of combined use TB LAM and Xpert assay.

Line 307-308: It is not clear how the values of 2 (33%) and 1 (17%) were obtained. Please add in the result section to clearly state how these values were obtained.

Line 320: It is not clear how the sensitivity of L-J culture (62.2%) was obtained. Please specify.

6. PLOS authors have the option to publish the peer review history of their article (what does this mean?). If published, this will include your full peer review and any attached files.

Reviewer #1: No

---

## [Author Response · Author response to Decision Letter 0]

4 Sep 2021

Response to Editor and Reviewers

We appreciate the editor and reviewers for the constructive comments which we have used to improve the quality of the manuscript. We have re-written some portions of the manuscript accordingly. We have carefully addressed the comments line by line as follows.

Response to Editor’s comments:

Comment 1: Please ensure that your manuscript meets PLOS ONE's style requirements, including those for file naming

Response 1: Thank you for your comment. We have accepted your comments and correction has made accordingly.

Comment 2: Please provide additional details regarding participant consent. In the ethics statement in the Methods and online submission information, please ensure that you have specified whether consent was informed.

Response 2: Written informed consent was obtained from each participants before enrolling into our study. This was described method parts. 

Comment 3: We note that you have indicated that data from this study are available upon request. PLOS only allows data to be available upon request if there are legal or ethical restrictions on sharing data publicly. If there are ethical or legal restrictions on sharing a de-identified data set, please explain them in detail and who has imposed them. Please also provide contact information for a data access committee, ethics committee, or other institutional body to which data requests may be sent. If there are no restrictions, please upload the minimal anonymized data set necessary to replicate your study findings as either Supporting Information files or to a stable, public repository and provide us with the relevant URLs, DOIs, or accession numbers.

Response 3: Thank you for your feedback. There is no legal restriction to share our data set. We uploaded our data set as a supporting information file (S2 File). 

Comment 4: PLOS requires an ORCID iD for the corresponding author in Editorial Manager on papers submitted after December 6th, 2016. Please ensure that you have an ORCID iD and that it is validated in Editorial Manager.

Response 4: Thank you for your comment. This is the ORCID iD of the corresponding author https://orcid.org/0000-0003-4751-2225.

Comment 5: We note that you have included the phrase “data not shown” in your manuscript. Unfortunately, this does not meet our data sharing requirements. PLOS does not permit references to inaccessible data. We require that authors provide all relevant data within the paper, Supporting Information files, or in an acceptable, public repository. Please add a citation to support this phrase or upload the data that corresponds with these findings to a stable repository (such as Figshare or Dryad) and provide and URLs, DOIs, or accession numbers that may be used to access these data. Or, if the data are not a core part of the research being presented in your study, we ask that you remove the phrase that refers to these data. 

Response 5: We agree with reviewer’s comment and the phrase referring the missed data was deleted from the manuscript. 

Additional editor question 1: Line 188- 'Forty-nine (33.3%) of the presumptive EPTB cases had the classification of colitis'-On what basis was this classification made and what was the sample collected from these patients?

Response1: About 49(33.3%) of the presumptive EPTB cases were classified as colitis based on the site of infection and symptoms. The site of infection for these 49 presumptive EPTB cases were abdomen and almost all of them experienced abdominal discomfort and pain during specimen collection. The sample collected from them were peritoneal fluid. So, due to the site of infection, abdominal discomfort and pain symptoms we have classified them as colitis.

Additional editor question 2: Although 4 of the 23 HIV positive patients were bacteriologically confirmed, it is not clear how many were positive by Xpert MTB/RIF. What is meant by 50% in Line 211? Similarly, for Determine TB LAM assay.

Response 2: Thank you for your comment and question. Although not described in the document, out of 23 HIV positive cases 3 of them were positive by Xpert MTB/RIF. Out of 23 HIV positive cases only 21 provided urine specimen and 7 were positive by TB LAM. Out of 23 HIV positive cases 6 were positive for EPTB by composite reference standard (CRS) and out of these 6 cases, only 3 were positive by Xpert MTB/RIF. So, the sensitivity of Xpert MTB/RIF for HIV positive cases were 3/6 (50%). The detection rate and diagnostic performance of Xpert MTB/RIF, TB-LAM and their combination in HIV-positive and –negative patients is now described in the supporting information file (S1 File). 

Additional editor question 3: Line 320- How was the sensitivity of LJ culture derived to be 62.2%? What was the gold standard? I think that was not within the scope of the study.

Response 3: Of course, determining the sensitivity of LJ culture is not our primary objective. Previous studies have shown that LJ culture has suboptimal sensitivity for diagnosis of EPTB and there are a significant number of cases which are negative on LJ culture but clinically diagnosed as TB. Taking note of this, we proposed to determine the sensitivity of LJ culture using CRS (composite reference standard) as a gold standard. In our study, a total of 37 patients were classified as EPTB based on CRS, of whom only 23 were positive by LJ culture. Thus, the sensitivity of LJ culture was 23/37(62.2%). We kept this just as an additional information.

Additional editor comment 4: Fig 1- The flow chart could start with inclusion of 147 patients suspected of presumptive TB instead of 160 patients.

Response 4: Thank you for your comment. We have accepted your comment and it was corrected on the revised manuscript.

Response to Reviewers' comments

Reviewers' comment 1: In the current study the authors aim to investigate the diagnostic performance of combined use of Xpert MTB/RIF and TB LAM Ag test for detection of M. tuberculosis complex among EPTB patients in a medical center of Ethiopia. Considering the high prevalence of EPTB and association of HIV in Ethiopia, the current diagnostic approach provides the important insight for increased detection of EPTB cases in clinical settings. However, the current manuscript needs to address several issues before approval. 

Response 1: We thank the reviewer for the constructive feedback. We tried our best to address comments/questions raised by the editor and reviewers. 

Response to Reviewer #1 Major Comments

Comment 1: According to the manufacturer (Abbot Laboratories, Chicago, US), the test name is ‘Determine™ TB-LAM Ag’. It is suggested to use the exact test name in the Title of the manuscript. Throughout the text, authors should be consistent mentioning the test name. Examples of some discrepancies are: ‘Determine TB LAM test’ in Line-31, ‘TB LAM’ in Line 41, ‘Determine TB-LAM test’ in line 136, ‘TB-LAM assay’ in line 140 and so on. It is suggested to use ‘Determine™ TB-LAM Ag (TB-LAM)’ when appears first in the text, followed by ‘TB-LAM’ all through the text.

Response 1: Thank you for your comment. In the revised manuscript, this was corrected.

Comment 2: In the introduction section, please add one paragraph describing the other published data on the performances of the Xpert MTB/RIF assay and Determine TB-LAM assay for detection of EPTB cases.

Response 2: We agree with reviewer’s comment and more evidence on the performance of Xpert MTB/RIF and TB-LAM was included in the introduction section of revised version.

Comment 3: One of the EPTB samples as has been described inconsistently throughout the text. For example, in Table-1 as ‘pus (abscess)’, Table-2 and throughout the text as ‘Abscess’. According to Line-191, it appears that samples were obtained from ‘Lymph node TB’. It would be easier for the readers to understand if the term ‘Lymph node aspirate’ is used all through the manuscript instead of ‘Abscess’.

Response 3: Thank you for your comment. We agree with reviewer’s comment and it was corrected as lymph node aspirate in the revised manuscript.

Comment 4: Gastric aspirate is considered as an alternative specimen for detection of pulmonary TB when patients cannot expectorate the sputum. As this study aims to detect the EPTB cases, authors should exclude the two ‘gastric aspirate’ samples from the study and re-analyze the results and correct the statistics accordingly in different relevant Tables, Figures and throughout the text. 

Response 4: Thank you for the interesting feedback. As the reviewer mentioned, in most cases, gastric aspirate is an alternative specimen for detection of pulmonary TB when patients cannot expectorate the sputum. But, in our case, the 2 gastric aspirate specimens were taken from patients who were suspected of having disseminated tuberculosis that is gastro intestinal tuberculosis.

Comment 5: For Table-2 and Table-3: Authors should check carefully that the statistics are presented correctly. It seems that the value of sensitivity, specificity, PPV and NPV and their corresponding 95% CI are not accurate. As for example, in Table-2 (based on the calculation of Fig-1), for ‘Determine TB LAM test’: the sensitivity would be 34.8% with 95% CI ranging from 16.4-57.3, specificity would be 92.7% (86.7-96.6), PPV would be 47.1% (27.7-67.4), and NPV would be 88.5% (85.0- 91.2). Please check both table 2 and 3, and correct accordingly. Please also add the 95% CI for specificity of Xpert MTB/RIF test in both tables. Should use the value of 95% CI in a consistent format like 0.0% (0.0-0.0) instead of 0.0% (0.0%-0.0%) in the tables and text.

Response 5: The reviewer’s comments are well taken and were corrected in the revised manuscript. The sensitivity, specificity, positive predictive value and negative predictive value for Xpert MTB/RIF and Smear microscopy were calculated from 147 participants but for TB-LAM the participants were 126. 

Comment 6: For clarity and better understanding authors should include the number of samples detected or not detected by the index tests compared to the reference tests along with the value of sensitivity and specificity. As for example, in Table-2, for ‘Determine TB LAM test’ please write the Sensitivity: 34.8% (34.1-35.5); 8/23 instead of 34.8% (34.1-35.5), and for specificity: 92.7% (86.7-96.6); 115/124 instead of 92.7% (86.7-96.6). Please add the numbers in Table-2 and Table-3 and describe accordingly in the text.

Response 6: We agree with what the reviewer suggested. In the revised version of the manuscript, the number of samples tested positive or negative by index test compared to the reference standard along with the values of sensitivity, specificity and predictive values are mentioned.

Comment 7: Figure 2: please describe in the result section (by mentioning the numbers of samples) how the percentages of different tests were obtained between HIV positive and negative patients

Response 7: Details of the percentages obtained for Xpert MTB/RIF and TB-LAM tests between HIV-positive and –negative cases are well addressed in the revised version of the manuscript. The detection rate and diagnostic performance of TB-LAM, Xpert MTB/RIF and combination of TB-LAM and Xpert MTB/RIF between HIV-positive and -negative participants was also addressed in supporting file (S1 File).

Comment 8: Line 215-217: It is not clear how the sensitivities of 44.4% for Xpert and 30.6% for TB LAM were obtained. Please clarify. Please also mention the statistical methods in ‘Methods and Material’ section that was used to obtain the p value.

Response 8: There were 126 patients who provided both site-specific extrapulmonary specimen (for Xpert MTB/RIF) and urine specimen (for TB-LAM). Comparison of Xpert MTB/RIF and TB-LAM was done only for 126 samples. Out of the 126 cases, about 36 were positive for EPTB by composite reference standard (CRS). Xpert MTB/RIF detected only 16 EPTB cases out of the 36 EPTB positive cases. So, the sensitivity of Xpert MTB/RIF test to detect EPTB among 126 cases were 16/36(44.4%). TB-LAM test detects only 12 EPTB cases out of the total EPTB positive cases (36). So, the sensitivity of TB-LAM test to detect EPTB among 126 cases were 12/36(33.3%). The sensitivity of TB-LAM test to detect EPTB among 126 cases was corrected on the revised manuscript from 30.6% to 33.3%.

Comment 9: Recently, Xpert MTB/RIF Ultra and Fujifilm SILVAMP TB-LAM have been appeared with higher sensitivities for detection of tuberculosis. In the discussion section authors should add one paragraph about the future applicability of these two assays compared with the current findings of Xpert MTB/RIF assay and Determine TB LAM assay.

Response 9: We appreciate the reviewer for the insightful complement. Regarding the Xpert MTB/RFI Ultra, we have a bit discussed the potential added value of Ultra over the Xpert MTB/RIF particularly in paucibacillary samples in the second paragraph of discussion. As we have mentioned in the discussion, the better performance of Ultra over Xpert MTB/RIF is mainly due to its lower analytical limit of detection (LLD;15 CFU/mL for Ultra versus 100–120 CFU/mL for Xpert MTB/RIF). Somewhere in the discussion (at the end of 4th paragraph), we also recommend the potential use of Xpert Ultra, when Xpert MTB/RIF yielded a negative result in body fluids such as pleural and peritoneal fluids. Regarding, the diagnostic yield of FujiLAM over TB-LAM, we also entertained this in the seventh paragraph of discussion as shown below’

“In their recent study, Kerkhoff et al. reported substantially higher sensitivity of Fujifilm SILVAMP TB-LAM (FujiLAM) over TB-LAM for detecting EPTB in HIV inpatients with moderate sensitivity in pleural fluid and CSF. This suggests a potential use of FujiLAM as a first-line test for the rapid detection of EPTB in HIV patients, with substantial added benefit in paucibacillary diseases such as pleural TB and TB meningitis.’’ 

Comment 10: Line 344: Please clarify the statement ‘equivalent to culture to diagnose EPTB’. It is not clear how the sensitivity of combined use of Xpert MTB/RIF and TB LAM was equivalent to ‘culture’ as the authors did not mention anything about the sensitivity of ‘culture’ in the manuscript.

Response 10: Thank you for your comment. In our study, LJ culture was positive for 23 cases out of the total 147 presumptive EPTB cases. Composite reference standard (CRS) detected 37 EPTB cases out of the total 147 presumptive EPTB cases. So, the sensitivity of L-J culture was 62.2% (23/37) that is equivalent to the previously calculated sensitivity of combined use of Xpert MTB/RIF and TB LAM test (61.1%). In the revised manuscript, we have mentioned the sensitivity of LJ culture.

Comment 11: While the text is readable, there are some grammar mistakes. Please correct the text to improve the comprehensibility.

Response 11: Thank you for your comment. We have thoroughly gone through the manuscript and edited it for grammar mistakes. 

Response to Reviewer #1 Minor Comments

Line 37:

Comment 1: The authors should mention the name of ‘reference’ test that is the ‘CRS’ to whom the results of Xpert MTB/RIF were compared.

Response 1: We have accepted your comment and it was mentioned in the revised manuscript.

Line 42:

Comment 2: Please write ‘EPTB cases’ instead of ‘TB cases’

Response 2: Comment accepted and corrected in the revised manuscript.

Line 41-42: 

Comment 3: Please correct and re-phrase the sentence ‘The combination of Xpert MTB/RIF and TB LAM detected 61.1%of all EPTB participants and 83.3% of HIV co-infected TB cases’ by mentioning the sensitivity instead of detection.

Response 3: The reviewers’ comments are noted and corrected in the revised manuscript.

Line 98: 

Comment 4: Should write Lowenstein Jensen (L-J) as it appears first in the text.

Response 4: Comment accepted and corrected on the revised manuscript.

Line 135: 

Comment 5: Please mention the volume of urine used for centrifugation.

Authors should specify why the LAM test was performed on refrigerated urine samples instead of freshly collected samples. Should discuss the point in the discussion section whether the sensitivity of LAM test varies between refrigerated versus fresh urine sample.

Response 5: As it has been mentioned in the Materials and Methods section, under the subheading of study subjects, sample collection and laboratory tests, all participants were requested to submit 10-30ml of urine sample, which was immediately refrigerated upon arrival to Mycobacteriology Research Center. Since there was a delay in the shipment process of TB-LAM kit, in our study TB-LAM test was done on the refrigerated urine samples. Though not reproduced by other researchers, Peter et al. has reported that the use of frozen urine has been associated with reduced TB- LAM sensitivity. We have mentioned the use of frozen urine sample as the possible limitation of our study and we have indicated that the low sensitivity of TB-LAM in our study could be partly because of the fact that the test was done on frozen urine sample. 

Line172: 

Comment 6: Please write as 35 years (IQR, 22-45).

Response 6: Comment noted and corrected on the revised manuscript.

Line-187: 

Comment 7: ‘Forty-nine (33.3%) of the presumptive EPTB cases had the classification of colitis’. Please clarify whether all of the 49 cases from where peritoneal fluid were collected had ‘colitis’

Response 7: Thank you for your comment. Out of 147, about 49(33.3%) of the presumptive EPTB cases were classified as colitis based on the site of infection and symptoms. Peritoneal fluid was collected from all (49) cases and all of them experienced abdominal discomfort and pain during specimen collection so that all of them were diagnosed as colitis.

Line-195 and 202: 

Comment 8: Authors should mention ‘detection rate’ instead of ‘prevalence’. Authors have mentioned in Table-1 that one sample among ‘Others’ group (2 gastric aspirates and 2 synovial fluids) was diagnosed as ‘Confirmed TB’. Please, specifically mention the sample type (either the gastric aspirate or synovial fluid) that was positive for TB in the table and add the description in line-196.

Response 8: We also agree with reviewers’ comments and considered in the revised version of the manuscript. 

Line 228-229: 

Comment 9: please write as (0.0%; 95% CI, 0.0-0.0) instead of ‘(30.6% (95%CI, 30%- 229 31%))’ and ‘(93.3% (95%CI, 93.2%-93.5%))’.

Response 9: We agree with reviewer’s suggestion and corrected accordingly. 

Line 230 and 238: 

Comment 10: Authors should include the ‘p value’ of the significant difference, and the test used.

Response 10: Comment well accepted and corrected in the revised manuscript.

Line 235-236: 

Comment 11: Please rephrase the sentence ‘Adding Determine TB LAM test to Xpert MTB/RIF test increases the sensitivity of Xpert MTB/RIF test from 43.2% to 61.1%’ as it does not increase the sensitivity of Xpert MTB/RIF test, but increases the sensitivity of combined use TB LAM and Xpert assay.

Response 11: Thank you for the interesting comment. We have now rephrased the sentence as “The combined use of TB- LAM and Xpert MTB/RIF tests had the sensitivity of 61.1% when compared to CRS, which was significantly higher than the sensitivity of each test alone” 

Line 307-308:

Comment 12: It is not clear how the values of 2 (33%) and 1 (17%) were obtained. Please add in the result section to clearly state how these values were obtained.

Response 12: We also agree with reviewer’s feedback. We found that it is better to describe the diagnostic performance of Xpert MTB/RIF, TB-LAM and their combined use among HIV-positive and negatives case, and we added one subheading in the result section. By including this, we assumed we have addressed some of questions/concerns raised from the reviewer.

Line 320: 

Comment 13: It is not clear how the sensitivity of L-J culture (62.2%) was obtained. Please specify.

Response 13: We have already addressed this question above in Response 10 under Major comments. To clarify it once again, of the total 37 EPTB cases diagnosed by CRS (the reference standard method used in our study), 23 were identified as EPTB positive by L-J culture. Using CRS (definite and probable-EPTB cases) as a gold standard, the sensitivity of L-J culture was 62.2% (23/37).

---

## [Decision Letter · Decision Letter 1]

1 Dec 2021

PONE-D-21-10435R1Combination of Xpert® MTB/RIF and DetermineTM TB-LAM Ag improves the diagnosis of extrapulmonary tuberculosis at Jimma University Medical Center, Oromia, EthiopiaPLOS ONE

Dear Dr. Tadesse,

Thank you for submitting your manuscript to PLOS ONE. After careful consideration, we feel that it has merit but does not fully meet PLOS ONE’s publication criteria as it currently stands. Therefore, we invite you to submit a revised version of the manuscript that addresses the points raised during the review process.

We look forward to receiving your revised manuscript.

Kind regards,

Shampa Anupurba, MD

Academic Editor

PLOS ONE

Journal Requirements:

Reviewers' comments:

Reviewer's Responses to Questions

**Comments to the Author**

1. If the authors have adequately addressed your comments raised in a previous round of review and you feel that this manuscript is now acceptable for publication, you may indicate that here to bypass the “Comments to the Author” section, enter your conflict of interest statement in the “Confidential to Editor” section, and submit your "Accept" recommendation.

Reviewer #1: (No Response)

Reviewer #2: All comments have been addressed

2. Is the manuscript technically sound, and do the data support the conclusions?

Reviewer #1: Yes

Reviewer #2: Yes

3. Has the statistical analysis been performed appropriately and rigorously? 

Reviewer #1: Yes

Reviewer #2: Yes

4. Have the authors made all data underlying the findings in their manuscript fully available?

Reviewer #1: Yes

Reviewer #2: Yes

5. Is the manuscript presented in an intelligible fashion and written in standard English?

Reviewer #1: Yes

Reviewer #2: Yes

6. Review Comments to the Author

Reviewer #1: It is appreciated that the authors have responded well the comments. However, it would be easier for the reviewer to go through the text if the specific line numbers are mentioned against the comments (where appropriate) in the cleaned version.

From Table 2 it is evident that TB-LAM test was compared against 36 and Xpert MTB/RIF test against 37 positive samples found by CRS. Would be easier for the reader if it is explained in short in the result section how these CRS have been found.

Line 227: the author should write ‘sensitivity’ instead of ‘pooled sensitivity’.

Line 179-180: ‘MedCalc Software Ltd (Comparison of proportions calculator) was used……….’ this sentence can be written as ‘Comparison of proportion between the methods was done by Chi square test using the ‘MedCalc Software’ (please include the weblink).

Reviewer #2: 1. The referencing style need to be uniform throughout the manuscript in accordance with Journal’s requirement. For example, please note the discrepancy in citing reference no. 1, 7, 13, 20, 24.

2. In ABSTRACT

Line 29 of the revised manuscript: Replace “examined for tuberculosis (TB)” with “tested for Mycobaterium tuberculosis complex (MTBC)”.

Line 40 of the revised manuscript: Write “and 100% specificity” instead of “with 100% specificity”.

3. In BACKGROUND

Line 60 of the revised manuscript: Replace “category III biosafety level” with “bio-safety level III facilties”.

4. In MATERIALS AND METHODS

Line 106-107 of the revised manuscript: There is no mention about collection of ‘gastric aspirate’ and ‘synovial fluid’.

Line 121-122 of the revised manuscript: Was decontamination also carried out for gastric aspirate specimens?

Line 179-180 of the revised manuscript: Earlier minor comment no. 10 by the reviewer is not addressed adequately. The tool used in determining the statistical difference haS been mentioned but the statistical test used for the same has not been mentioned in the revised version of the manuscript.

5. In RESULTS

Line 206-107 of the revised manuscript: With respect to addition editor’s question no. 1 and the reply by the authors, will it be better if the authors mention the ‘colitis’ cases as ‘suspected abdominal TB’ cases because abdominal pain and discomfort could be due to causes other than colitis?

Line 215-216 of the revised manuscript: “1 disseminated TB from gastric aspirate”- It remains dubious how detecting MTBC from gastric aspirate only can label the case as ‘disseminated TB’. Clarification is needed.

Line 242-243 of the revised manuscript: It is not clear whether culture confirmation (23 patients) could only be done only among 126 presumptive EPTB cases who submitted urine samples for TB-LAM. In other words, was there no culture confirmed cases among the other 21 presumptive EPTB cases who did not submit urine sample for TB-LAM?

This needs to be clarified.

6. In DISCUSSION

Line 316 of the revised manuscript: Replace ‘Additionally’ with ‘However’.

Line 355 of the revised manuscript: Write ‘End TB strategy’ instead of ‘end TB strategy’.

7. PLOS authors have the option to publish the peer review history of their article (what does this mean?). If published, this will include your full peer review and any attached files.

Reviewer #1: No

Reviewer #2: **Yes: **Arghya Das, MD

---

## [Author Response · Author response to Decision Letter 1]

28 Dec 2021

Response to Reviewers

We appreciate the reviewers for the constructive comments which we have used to improve the quality of the manuscript. As we did in the last time, we have re-written some portions of the manuscript accordingly. We have also carefully addressed the reviewers’ comments line by line as follows:

Response to Reviewers' comments 

Reviewer 1

Comment 1: It is appreciated that the authors have responded well the comments. However, it would be easier for the reviewer to go through the text if the specific line numbers are mentioned against the comments (where appropriate) in the cleaned version. 

Response 1: Sorry for not mentioning the specific line numbers against the comments in the last cleaned version. We agree with the reviewer’s comment. Now, in the recent version, the specific line numbers are indicated against the comments in the cleaned version. 

Comment 2: From Table 2, it is evident that TB-LAM test was compared against 36 and Xpert MTB/RIF test against 37 positive samples found by CRS. Would be easier for the reader if it is explained in short in the result section how these CRS have been found.

Response 2: In general, as indicated in the method section (Line 166-169 ), CRS, which comprises of smear microscopy, L-J culture and clinical improvement after ATT initiation, was used as gold standard to calculate the sensitivity, specificity and predictive values of both TB-LAM and Xpert MTB/RIF. Specifically for calculation of sensitivity, as it was stated by the reviewer, TB-LAM test was compared against 36 CRS-positive cases whereas Xpert MTB/RIF was compared against 37 CRS-positive cases. This is now addressed in the revised version of the manuscript. See Line224-226 of cleaned version of the manuscript.

Comment 3: Line 227: the author should write ‘sensitivity’ instead of ‘pooled sensitivity’

Response 3: We have modified it accordingly. Shown in Line 232.

Comment 4: Line 179-180: ‘MedCalc Software Ltd (Comparison of proportions calculator) was used……….’ this sentence can be written as ‘Comparison of proportion between the methods was done by Chi square test using the ‘MedCalc Software’ (please include the weblink).

Response 4: We agree with the reviewer’s feedback and we have modified it accordingly as shown Line 177-180 of cleaned version of the manuscript .

Reviewer 2

Comment 1: The referencing style need to be uniform throughout the manuscript in accordance with Journal’s requirement. For example, please note the discrepancy in citing reference no. 1, 7, 13, 20, 24

o Response 1: The reviewer’s comment is accepted and corrected accordingly. 

Comment 2: In ABSTRACT;

Comment 2.1: Line 29 of the revised manuscript: Replace “examined for tuberculosis (TB)” with “tested for Mycobacterium tuberculosis complex (MTBC)”. 

o Response 2.1: Comment accepted and corrected accordingly. Shown in Line 27 of cleaned version.

Comment 2.2: Line 40 of the revised manuscript: Write “and 100% specificity” instead of “with 100% specificity”. 

o Response 2.2: Comment accepted and corrected accordingly. Shown in Line 38 of cleaned version.

Comment 3: In BACKGROUND

Comment 3.1: Line 60 of the revised manuscript: Replace “category III biosafety level” with “bio-safety level III facilities”.

o Response 3: We agree with the reviewer’s suggestion and corrected accordingly. Shown in Line 58 of cleaned version.

Comment 4: In MATERIALS AND METHODS 

Comment 4.1: Line 106-107 of the revised manuscript: There is no mention about collection of ‘gastric aspirate’ and ‘synovial fluid’. 

o Response 4.1: Gastric aspirate and synovial fluid samples were collected as per the standard of care. Information on the collection of synovial fluid and gastric aspirate is shown in Line 104-105 of cleaned version.

Comment 4.2: Line 121-122 of the revised manuscript: Was decontamination also carried out for gastric aspirate specimens? 

o Response 4.2: Of course, like that of lymph node aspirates and blood stained samples, gastric aspirate samples were decontaminated by the standard N-acetyl-L-cysteine and sodium hydroxide (NALC/NaOH) to reduce overgrowth from contaminating microorganism. This was addressed in the revised cleaned version. See Line 119 of cleaned version

Comment 4.3: Line 179-180 of the revised manuscript: Earlier minor comment no. 10 by the reviewer is not addressed adequately. The tool used in determining the statistical difference has been mentioned but the statistical test used for the same has not been mentioned in the revised version of the manuscript.

o Response 4.3: As we have explained in the revised version (Line 177-180), presence or absence of statistically significant difference is determined based on the P-value. The P-value<0.05 showed statistically significant difference whereas the P-value>0.05 showed no difference (similar, not merely the same). 

Comment 5: In RESULTS

Comment 5.1: Line 206-107 of the revised manuscript: With respect to addition editor’s question no. 1 and the reply by the authors, will it be better if the authors mention the ‘colitis’ cases as ‘suspected abdominal TB’ cases because abdominal pain and discomfort could be due to causes other than colitis?

o Response 5.1: We agree with the feedback from the reviewer and corrected it. See Line 206-207 of cleaned version.

Comment 5.2: Line 215-216 of the revised manuscript: “1 disseminated TB from gastric aspirate”- It remains dubious how detecting MTBC from gastric aspirate only can label the case as ‘disseminated TB’. Clarification is needed.

o Response 5.2: This is an interesting observation from the reviewer. We strongly agree with the reviewer’s concern and one MTBC case from gastric aspirate cannot signify disseminated TB. However, pulmonary manifestation was excluded from this patient and we assumed this patient had extrpulmonary form of TB. Accordingly, we revised the manuscript as shown in Line 215-217 of cleaned version.

Comment 5.3: Line 242-243 of the revised manuscript: It is not clear whether culture confirmation (23 patients) could only be done only among 126 presumptive EPTB cases who submitted urine samples for TB-LAM. In other words, was there no culture confirmed cases among the other 21 presumptive EPTB cases who did not submit urine sample for TB-LAM? This needs to be clarified.

o Response 5.3: This is again another interesting comment from the reviewer. The other reviewer also raised similar concern. There was one culture-positive case from presumptive cases who did not submit urine sample. That could be why we excluded this case from CRS when determining the diagnostic accuracy of TB-LAM. As it was shown in Table 2, the TB-LAM test was compared against 36 CRS positive cases whereas Xpert MTB/RIF test was compared against 37 CRS positive cases. For combination of TB-LAM and Xpert MTB/RIF, the 36 cases who have all the three test results were considered. Shown in Line 224-226 of cleaned version.

Comment 6: In DISCUSSION

Comment 6.1: Line 316 of the revised manuscript: Replace ‘Additionally’ with ‘However’.

o Response 6.1: We agree. This is corrected in the revised version of the manuscript as shown in Line 315.

Comment 6.2: Line 355 of the revised manuscript: Write ‘End TB strategy’ instead of ‘end TB strategy’.

o Response 6.2: It is written as End TB strategy (Line 358) in the revised version of the manuscript. 

Comment 7: PLOS authors have the option to publish the peer review history of their article (what does this mean?). If published, this will include your full peer review and any attached files.

Response 7: We are OK with the option to publish the peer review history.

---

## [Decision Letter · Decision Letter 2]

14 Jan 2022

Combination of Xpert® MTB/RIF and DetermineTM TB-LAM Ag improves the diagnosis of extrapulmonary tuberculosis at Jimma University Medical Center, Oromia, Ethiopia

PONE-D-21-10435R2

Dear Dr. Tadesse,

We’re pleased to inform you that your manuscript has been judged scientifically suitable for publication and will be formally accepted for publication once it meets all outstanding technical requirements.

Kind regards,

Shampa Anupurba, MD

Academic Editor

PLOS ONE

Additional Editor Comments (optional):

Reviewers' comments:

Reviewer's Responses to Questions

**Comments to the Author**

1. If the authors have adequately addressed your comments raised in a previous round of review and you feel that this manuscript is now acceptable for publication, you may indicate that here to bypass the “Comments to the Author” section, enter your conflict of interest statement in the “Confidential to Editor” section, and submit your "Accept" recommendation.

Reviewer #1: All comments have been addressed

Reviewer #2: All comments have been addressed

2. Is the manuscript technically sound, and do the data support the conclusions?

Reviewer #1: Yes

Reviewer #2: Yes

3. Has the statistical analysis been performed appropriately and rigorously? 

Reviewer #1: Yes

Reviewer #2: Yes

4. Have the authors made all data underlying the findings in their manuscript fully available?

Reviewer #1: Yes

Reviewer #2: Yes

5. Is the manuscript presented in an intelligible fashion and written in standard English?

Reviewer #1: Yes

Reviewer #2: Yes

6. Review Comments to the Author

Reviewer #1: (No Response)

Reviewer #2: (No Response)

7. PLOS authors have the option to publish the peer review history of their article (what does this mean?). If published, this will include your full peer review and any attached files.

Reviewer #1: No

Reviewer #2: **Yes: **Arghya Das, MD

---

## [Editor Report · Acceptance letter]

25 Jan 2022

PONE-D-21-10435R2 

Combination of Xpert MTB/RIF and Determine^TM^ TB-LAM Ag improves the diagnosis of extrapulmonary tuberculosis at Jimma University Medical Center, Oromia, Ethiopia 

Dear Dr. Tadesse:

I'm pleased to inform you that your manuscript has been deemed suitable for publication in PLOS ONE. Congratulations! Your manuscript is now with our production department. 

Kind regards, 

on behalf of

Dr. Shampa Anupurba 

Academic Editor

PLOS ONE